# P-21 Activated Kinases in Liver Disorders

**DOI:** 10.3390/cancers15020551

**Published:** 2023-01-16

**Authors:** Xun Qiu, Hanzhi Xu, Kai Wang, Fengqiang Gao, Xiao Xu, Hong He

**Affiliations:** 1Key Laboratory of Integrated Oncology and Intelligent Medicine of Zhejiang Province, Department of Hepatobiliary and Pancreatic Surgery, Affiliated Hangzhou First People’s Hospital, Zhejiang University School of Medicine, Hangzhou 310006, China; 2Zhejiang University School of Medicine, Hangzhou 310058, China; 3Westlake Laboratory of Life Sciences and Biomedicine, Hangzhou 310024, China; 4Key Laboratory of Integrated Oncology and Intelligent Medicine of Zhejiang Province, Hangzhou 310006, China; 5Department of Surgery, University of Melbourne, Austin Health, 145 Studley Rd., Heidelberg, VIC 3084, Australia

**Keywords:** p21 Activated Kinase, liver cancer, hepatitis, liver fibrosis, hepatic ischemia-reperfusion injury, small molecular inhibitors

## Abstract

**Simple Summary:**

Liver disorders, particularly liver cancer and hepatitis, are considered major public health challenges. Nearly 90% of patients with liver cancer are diagnosed at late stages of the disease, which reduces the treatment options, having a significant negative impact on patient survival. Hepatitis is one of major risk factors of liver cancer. The p21 Activated Kinases (PAKs) play important roles in various diseases, and thus become targets for therapeutical purpose. Their roles in liver disorders are emerging. This review highlights the importance of alteration of PAKs within liver disorders, emphasizes their effects on liver cancer and hepatitis, and discusses the potential therapeutical effect of PAKs in liver disorders. We aim to provide an overview on current progress of PAKs in liver diseases, and a useful guide to research into PAK inhibitors and the potential application.

**Abstract:**

The p21 Activated Kinases (PAKs) are serine threonine kinases and play important roles in many biological processes, including cell growth, survival, cytoskeletal organization, migration, and morphology. Recently, PAKs have emerged in the process of liver disorders, including liver cancer, hepatic ischemia-reperfusion injury, hepatitis, and liver fibrosis, owing to their effects in multiple signaling pathways in various cell types. Activation of PAKs promotes liver cancer growth and metastasis and contributes to the resistance of liver cancer to radiotherapy and chemotherapy, leading to poor survival of patients. PAKs also play important roles in the development and progression of hepatitis and other pathological processes of the liver such as fibrosis and ischemia-reperfusion injury. In this review, we have summarized the currently available studies about the role of PAKs in liver disorders and the mechanisms involved, and further explored the potential therapeutic application of PAK inhibitors in liver disorders, with the aim to provide a comprehensive overview on current progress and perspectives of PAKs in liver disorders.

## 1. Introduction

The p21 Activated Kinases (PAKs) are serine-threonine kinases containing six members, and play important roles in a variety of biological processes such as cell growth, survival, cytoskeletal organization, migration, and morphology [1]. These kinases act downstream of the Ras-related Rho GTPases cell division control protein 42 (CDC42) and RAC, and are involved in multiple signaling pathways such as phosphoinositide 3-kinase (PI3K)/protein kinase B (AKT) and Wnt/β-catenin pathways [2,3]. PAKs can be divided into two groups based on structural and biological properties. Group I PAKs include PAK1-3, and group II PAKs contain PAK4-6. PAKs have been involved in many physiological and pathological processes. Their roles in cancer have been extensively investigated. Increasing evidence implicates PAK effects in liver disorders including liver cancer, hepatitis, liver fibrosis, and hepatic ischemia-reperfusion injury (IRI) [4,5,6,7]. In this review, we will discuss the role of PAKs in liver disorders with a focus on liver cancer and hepatitis and the underlying mechanisms, and we will explore the potential therapeutic effects of PAKs in liver disorders. The upstream regulators of PAKs and the roles of PAKs in liver disorders are summarized in Figure 1 and Table 1, respectively. 

## 2. Structure and Activation of PAKs Family

PAK proteins contain a kinase domain at the C-terminal, and an autoinhibitory domain (AID) and a GTPase-binding domain (GBD) at the N-terminus [8]. The AID of Group I PAKs overlaps with the GBD and binds in trans to the kinase domain of another PAK molecule to form a dimer, preventing the formation of an activation loop. The binding of Rho GTPases or other proteins to the GBD dissociates the binding of AID with the kinase domain and causes a conformational change, which in turn triggers autophosphorylation of Ser 144 and Thr 423 and activation of the kinase. Group II PAKs function as monomers and they have AID-like domains different from the AID in group I PAKs [9]. There are two proposed models to explain the activation of group II PAKs. One model is the AID-like domain, which binds to the kinase domain in cis, forming an inactive conformation. When CDC42 or RAC binds to GBD, this conformation will change and release the kinase domain from auto-inhibition [10]. The other model contains two phases for activation. In addition to the binding of Rho GTPases, it also requires the involvement of proteins that contain SH3 domains, which interact with the AID-like domains [11] (Figure 2).

PAKs act in multiple signaling pathways that are crucial for physiological and pathological processes. Particularly, overexpression of PAKs in many types of cancers is correlated with cancer progression and therapeutical resistance. Both PAK1 and PAK4 can activate the PI3K/AKT pathway, one of the key pathways for cell growth and survival. PAKs promote chemoresistance of cancer cells through activating the PI3K/AKT pathway [12]. PAK1 stimulates cell migration by activation of AKT [13]. PAK4 bind to p85, the regulatory subunit of PI3K, which in turn increases the activity of PI3K and phosphorylation of AKT, thereby promoting the migration of cancer cells [14]. On the other hand, PI3K/AKT can also regulate the activation of PAKs. It has been suggested that a positive feedback loop may exist between PAKs and PI3K/AKT in the progression of cancers [15,16,17]. RAS/RAF/mitogen-activated protein kinase (MEK)/extracellular-regulated kinase (ERK) axis is a well-known pathway contributing to the growth, survival, and metabolism of cancer cells. PAKs can phosphorylate MEK and RAF at Ser298 and at Ser338, respectively, promoting the activation of RAS-mediated RAF/MEK/ERK signaling [18]. In IκB/nuclear factor-κB (NF-κB) signaling, PAK1 promoted the phosphorylation and degradation of IκB, leading to the activation and nuclear translocation of NF-κB [19]. Blockading PAK1-activated NF-κB inhibited the growth of HCC and alleviated the inflammatory response in acute pancreatitis [20,21]. PAKs can activate β-catenin signaling and stabilize the activation of β-catenin, contributing to the chemotherapy resistance of cancer [22,23,24]. PAKs can form a complex with β-catenin in the nucleus, which can further stabilize β-catenin. PAK4 can phosphorylate β-catenin at Ser 675, preventing the ubiquitination and degradation of β-catenin [25,26]. PAKs regulate the activation of Lin-11/Isl-1/Mec-3 kinase and matrix metalloproteinases, which are associated with cancer cell metastasis [27,28]. 

Except for their roles in cancers, alterations of PAKs are also implicated in other human diseases. By regulation of cytoskeleton dynamics, PAKs play key roles in brain development and thus neurological disorders [29]. Mutations in PAK1 are identified in intellectual disability associated with seizures and speech delay whereas mutations in PAK2 and PAK3 are found in autism and X-linked mental retardation. PAK1 regulates the phosphorylation of the proteins controlling heart rate and thus plays an important role in cardiac diseases [30]. PAK2 and PAK4 have been implicated in the dysfunction of pancreatic acinar [31]. PAK-mediated activation of Wnt and G-protein pathways have been reported in type 2 diabetes [32].

## 3. PAKs in Liver Cancers

Liver cancer is the sixth most common cancer worldwide and the second most common cause of cancer-related death. Hepatocellular carcinoma (HCC) accounts for 85% of liver cancer [33,34]. The major risk factors for liver cancer are chronic infection with hepatitis B virus (HBV) or hepatitis C virus (HCV) [34,35,36]. PAKs have been reported to be upregulated in many cancers including breast cancer, glioblastoma, pancreatic cancer, and HCC, promoting tumor growth and metastasis [7,37,38]. Increased PAKs expression endowed tumors with chemoresistance, and thus PAKs become attractive therapeutical targets for cancer treatment and biomarkers to predict prognosis [39,40,41]. The roles of PAKs in liver cancer, especially HCC, are discussed below (Figure 3).

### 3.1. Roles of Group I PAKs in Liver Cancer

PAK1 has been the most investigated among the I PAKs group. Overexpression of PAK1 was most frequently observed in HCC and associated with the growth and metastasis of HCC. PAK1 affects HCC progression via multiple pathways, PAK1 stimulated proliferation and inhibited apoptosis of HCC cells by activation of β-catenin signaling, and facilitated migration and invasion by upregulation of Snail [42]. Genetic inhibition of PAK1 suppressed xenograft tumor growth through Snail-, β-catenin-, and p53/p21-mediated pathways [42,43]. Treatment of phorbol ester increased PAK1, which in turn promoted HCC metastasis by activation of paxillin and c-Jun N-terminal kinase (JNK) [44]. PAK1 also promoted HCC metastasis by regulating the formation and degradation of actin stress fibers [45]. PAK1 enhanced vasculogenic mimicry in HCC through the stabilization of hypoxia-inducible factor-1α and phosphorylation of Vimentin [46]. Activation of PAK1-mediated epidermal growth factor receptor (EGFR) and vascular endothelial growth factor receptor (VEGFR) pathways inhibited anoikis, a form of programmed cell death, contributing to the anoikis resistance of HCC cells [47,48].

A natural product, resveratrol, inhibited HCC cell proliferation and survival by downregulation of PAK1, cyclin D1, mitogen-activated protein kinase (MAPK), and AKT [49]. In a liver carcinogenesis animal model, diethylnitrosamine-induced HCC progression was correlated with increased levels of PAK1, cyclin D1, MAPK, and AKT [35]. Myricetin induces apoptosis of HCC cells by abrogating RAS-activated PAK1 and inhibiting MAPK/ERK and PI3K/AKT signaling [50]. *KRAS* or *BRAF* mutations have been shown to promote hepatic vascular cavernomas [51]. Although PAK1 has been reported to affect the activation level of RAS/MEK/ERK signaling [52,53], the role of PAK1 in hepatic vascular cavernomas is unknown. The Smad4-MYO18A-PP1A complex has been reported to inhibit the proliferation, migration, invasion, and resistance to pemigatinib of cholangiocarcinoma through dephosphorylating PAK1 and suppressing subsequent intranuclear localization of β-catenin [22].

MicroRNAs (miRNAs) are small, non-coding RNAs that bind to specific mRNA targets to regulate gene expression post-transcriptionally. Altered expression of miRNAs is associated with liver metabolic dysfunction and tumor development. LINC00460, a long intergenic non-protein coding RNA, promoted HCC progression by stimulation of PAK1 through suppressing miR-485-5p [54]. Zhou and colleagues also reported that miR-15b was targeted by PAK1 to promote the proliferation, migration, and invasion of HCC cells [55].

HBV infection is a well-recognized risk factor for liver cancer. Induction of hepatitis B virus X protein to HCC cells promoted anchorage-independent growth and anoikis resistance by upregulation of PAK1, and knockdown of PAK1 inhibited tumor growth of HCC cells and their resistance to anoikis. More importantly, PAK1 expression is positively correlated to HBV infection, worse prognosis, and portal vein tumor thrombosis in patients [56]. 

PAK2 played a critical role in HCC metastasis. MiR-26a inhibited metastasis of HCC through downregulation of PAK2 [57]. Sato et al. reported that PAK2 phosphorylation was associated with AKT activation in transforming growth factor-β (TGF-β)-induced migration of HCC cells [58]. PAK3 is mainly expressed in the central nervous system for regulation of neuronal plasticity and spinogenesis [59]. However, Gao and colleagues recently found that PAK3 was upregulated in HCC and stimulated the epithelial mesenchymal transition (EMT) of tumor cells through activating the TGF-β/Smad pathway [60].

### 3.2. Roles of Group II PAKs in Liver Cancer

PAK4 enhanced HCC cell survival by modulating caspase-8 and NF-κB pathways [61]. In an HCC transgenic mouse model, miR-199-3p inhibited HCC growth by targeting the PAK4/RAF/MEK/ERK pathway [62,63]. On the contrary, an increased cyclin-dependent kinase 5 (CDK5) regulatory subunit-associated protein 3 in HCC accelerated tumor metastasis bound to and activated PAK4 [64]. PAK4 promoted migration and invasion of HCC cells through direct phosphorylation of p53 at S215 [65]. MiR-1271 suppressed HCC growth and metastasis through downregulation of RAF/MEK/ERK signaling, achieved by inhibiting Zic2/PAK4. PAK4 was overexpressed in metastatic tumor tissues compared with primary tumor and normal tissues, and PAK4 expression was correlated with worse survival of HCC patients. The results from multivariate analyses of 615 HCC patients showed that PAK4 was an independent indicator of poor prognosis [66]. MiR-433 was reduced in HCC tissues. MiR-433 repressed the viability of HCC cells through directly binding to the sequence at 3’-UTR of PAK4 mRNA, which in turn inhibited PAK4 expression [67]. MiRNAs are key upstream regulators that directly modulate the translation of multiple oncogenic proteins including PAK4. A miRNA cocktail therapy targeting PAK4, mechanistic target of rapamycin (mTOR), and RAS homolog gene family member C (RHOC), showed remarkable anti-HCC effects in patient-derived xenografts [68].

Immunohistochemical results from tumor and corresponding para-tumor tissues of 273 HCC patients showed that PAK5 was overexpressed in tumor tissues and strongly correlated with tumor progression and worse prognosis of HCC patients [69]. MiR-129 and miR-526a, which were downregulated in HCC tissues and cell lines, have been shown to suppress the proliferation and invasion of HCC cells through interfering the translation of PAK5 [70,71]. Gene silencing of PAK5 attenuated the proliferation and colony formation of HepG2 cells and inhibited the tumor progression in vivo [72]. Overexpression of PAK5 reduced the apoptosis induced by cisplatin in HCC cells [73]. PAK5 conferred HCC with chemoresistance to adriamycin and 5-fluorouracil by phosphorylating β-catenin and promoting its nuclear translocation, leading to the increased transcription of adenosine triphosphate-binding cassette subfamily B member 1 (ABCB1), a gene responsible for multidrug resistance [69]. Recently, it has been reported that miR-138-1-3p enhanced the sensitivity of HCC to sorafenib by inhibiting PAK5/β-catenin/ABCB1 axis [24]. Inhibition of PAK5 also sensitized HCC to radiotherapy by inducing cell apoptosis, cell cycle arrest, and DNA damage [74].

Elevated PAK6 was also found in HCC tissues compared with para-tumor tissues, and PAK6 expression was positively associated with the high proliferative ability of HCC and poor prognosis of patients [75]. A recent study indicated PAK6-regulated mitosis, through downregulation of Eg5, could repress PAK6 expression and in turn, inhibit PAK6 in hepatocarcinogenesis [76]. Contrarily, Liu et al. reported that PAK6 played a tumor suppressive role by showing that HCC patients with lower PAK6 had a higher tumor node metastasis stage and worse survival [77]. Further studies are necessary to explore the role of PAK6 in HCC.

## 4. PAKs in Hepatic Ischemia/Reperfusion Injury

Hepatic IRI commonly occurs in the process of hemorrhagic shock, liver surgery, and liver transplantation [78,79]. The underlying mechanisms of hepatic IRI relate to the elevated oxidative stress and activation of immune-metabolic responses, which are harmful to normal cellular structures and functions, and result in liver damage [80]. PAKs participate in the modulation of immune responses and inflammation [81,82], and PAKs have been reported to play a critical role in hepatic IRI [6,83]. 

Neuregulin-1/PAK1 axis was found to decrease IRI of liver grafts with or without steatosis through increasing vascular endothelial growth factor-α and insulin growth factor-1 levels, respectively [83]. PAK4 was upregulated in hepatic IRI in mice and humans to promote hepatic hypoxia-reoxygenation-induced damage through phosphorylating nuclear factor erythroid 2-related factor 2 and suppressing its transcriptional activity [6]. Genetic knockout or pharmacological inhibition of PAK4 reduced the inflammation and necrosis of hepatocytes [6]. These results suggest that PAK4 inhibition may protect the liver from IRI-induced damage. The roles of other PAK members in liver IRI have not been investigated.

## 5. PAKs in Hepatitis

Hepatitis is defined as inflammation of the liver, and its etiologies include virus infection, parasitic invasion, alcohol abuse, autoimmunity imbalance, and metabolic disorder. The main characteristic of hepatitis is represented by the infiltration of inflammatory cells, which induces apoptosis and necrosis of hepatocytes leading to liver damage and dysfunctions [84,85,86,87]. The role of PAKs in hepatitis, especially virus-related ones, are discussed here.

In hepatitis B, phosphorylation of PAK1 induced the translocation of RAF-1 to mitochondrial, which in turn facilitated the anti-apoptotic effect of HBV X protein in hepatocyte [88]. In fact, HBV X protein can activate PAK1, which protected HCC from anoikis and promoted the metastasis of HCC [56]. Results from the PCR showed that the *PAK3* gene was the preferential integration site of HBV DNA and the integration potentially changed the expression of PAK3 [5]. Yu and colleagues conducted a prospective nested case-control study and analyzed the peripheral blood samples of enrolled 240 HBeAg seropositive chronic hepatitis B patients treated with interferon-α (IFN-α). They have found that patients with genotype *TT* of rs9676717 in *PAK4* gene were more sensitive to IFN-α therapy (*p* = 0.015) [89]. However, there were few associations between polymorphisms in *PAK4* gene and susceptibility to HBV-related HCC [90].

Different from the promoting role of PAK1 in HBV, PAK1 had potentially inhibited HCV replication, and its antiviral effects were independent of interferon regulatory factor 3 but dependent on the mTOR-activated PI3K/AKT and ERK [91]. HCV protected the infected hepatocytes from cell death by suppressing apoptosis and inflammatory reaction partially through the upregulation of PAK2 [92]. It was reported that quasi-enveloped mammalian hepatitis E virus infection relied on the regulation of the CDC42-associated PAK1/non-myosin IIA/Cofilin pathway, which regulated F-actin polymerization and membrane remodelling [93,94].

In a mouse model infected with *Schistosoma japonicum*, PAK1 in Kupffer cells was elevated and facilitated the differentiation of CD4^+^ T cells to T helper 17 cells via the NF-κB/interferon regulatory factor 1/interleukin-6 pathway, leading to aggravated hepatic inflammation [95]. Increased PAK1 expression in Kupffer cells was also noticed in patients with autoimmune hepatitis, and the PAK1 expression was found to be associated with disease progression [96]. The increased expression of PAK6 found in alcoholic hepatitis may contribute to tumorigenesis [97].

## 6. PAKs in Liver Fibrosis

Liver fibrosis is implicated in the interaction between injured hepatocytes, inflammatory cells, and hepatic myofibroblasts. In response to injury, hepatocytes will express more profibrogenic miRNAs and proteins such as TGFβ and Notch, thus initiating fibrosis [98]. The transformation from hepatic stellate cells to myofibroblasts that produce extracellular matrix (ECM) provides a protection to liver tissue from injury, and there is a balance between ECM production and degradation. However, the excessive accumulation of ECM produced by consistently activated myofibroblasts will impair the normal physiological structure and function of the liver, resulting in the occurrence of liver fibrosis [99]. 

PAK1 and PAK3 were upregulated in activated hepatic myofibroblasts and promoted fibrotic effects by enhancing the expression of integrin β-1, which is essential for myofibroblasts activation and ECM production [100]. Chen et al. reported that Kolotho expression was significantly associated with liver fibrosis in HCC patients, contributing to the resistance of HCC cells to anoikis by activating VEGFR2/PAK1 signaling [47]. Using a co-culture system and a CCl_4_-induced fibrosis mouse model, Zhou et al. revealed that human umbilical cord mesenchymal stem cells inactivated LX-2 cells and consequently reduced liver fibrosis through increasing miR-455-3p-mediated inhibition on PAK2 translation [4]. It was also reported that Atg7-deficient mice exhibited perisinusoidal/pericellular fibrosis in liver with age, and PAK4 may act as a crucial mediator in this pathological process [101].

## 7. Therapeutic Effects of PAK Inhibitors in Liver Disorders

The main strategy for treating HBV or HCV-related liver diseases is long-term administration of antiviral drugs such as entecavir, disoproxil, and IFN-α. Different from HCV, HBV has the characteristic of escaping innate immunity, thus immune therapy is also a vital treatment for HBV [102,103]. For metabolism-related liver diseases, limiting alcohol consumption, changing eating patterns and diet composition, and reducing insulin resistance and improving lipid metabolism are effective regimens [104,105].

Liver resection and transplantation are considered as curative treatments for HCC. However, only a small proportion of patients can have these surgical operations at the diagnosis stage and the recurrence after surgery is high. Most HCC patients are diagnosed at advanced stages of the disease and can only have non-surgical treatments, including chemo-, radio-, and immune-therapies. The first-line therapy for advanced HCC includes broad-spectrum tyrosine kinase inhibitors, such as sorafenib and lenvatinib, and combinational therapies of immunotherapy and anti-angiogenesis therapy [106]. Sorafenib and the newly approved lenvatinib are oral multi-tyrosine kinase inhibitors, and there was no significant difference between the two inhibitors in improving the overall survival of unresectable HCC patients (13.6 versus 12.3 months) [107]. The combination of immunotherapy of programmed cell death-ligand 1 (PD-L1) blockade and anti-angiogenesis therapy of VEGF blockade has been approved as one of best available first-line treatments [106]. Although these strategies provide an extension in survival, they cause broad spectrum toxic side effects, and patients eventually develop therapy resistance. To overcome therapy resistance and in searching for alternative treatment, gene therapy emerges as a promising remedy for HCC, especially with gene mutation [108].

PAKs play key roles in liver disorders and thus are considered useful targets for the treatment of liver diseases. PAK inhibitors have been developed and can be divided into two categories depending on the binding sites within PAK molecules, ATP-competitive inhibitors, and allosteric PAK inhibitors [109]. ATP-competitive PAK inhibitors block PAK phosphorylation by targeting the ATP-binding pocket within the kinase domain. This type of PAK inhibitors is further classified into aminopyrazole-based inhibitors, aminopyrimidine-based inhibitors, indolocarbazole-based inhibitors, 2-amino pyrido[2,3-d] pyrimidine-7(8H)-one-based inhibitors, and other ATP-competitive inhibitors. Compared with ATP-competitive PAK inhibitors, allosteric PAK inhibitors have the potential to be more selective and display discriminative inhibitory activity among PAK family proteins [110,111]. 

The studies of the effects of PAK inhibitors in liver disorders are limited (Figure 4). PAK4 was upregulated and played a critical role in the process of hepatic IRI. Application of an ATP-competitive PAK4 inhibitor ND201651 reduced hepatic IRI [6]. Another novel specific PAK4 inhibitor, SPA7012, a pyrazolo [3,4-d] pyrimidine derivative, also showed an anti-inflammatory effect on the liver during hepatic IRI [112]. More experimental evidence is necessary for its clinical application.

IPA-3 is a non-ATP-competitive allosteric inhibitor targeting PAK1 phosphorylation. Since group I PAKs share the same inhibitory domain in the N-terminal sequence, IPA-3 also suppresses the activation of PAK2 and PAK3 [113]. Administration of IPA-3 remarkably improved the bile duct ligation and CCl_4_-induced fibrosis in mice without hepatotoxicity [100]. IPA-3 suppressed HCC growth by inhibition of PAK1-mediated JNK phosphorylation and NF-κB intranuclear localization [21]. HCC patients with high Klotho expression showed a poor overall survival. IPA-3 could sensitize HCC cells to anoikis by blocking the Klotho/VEGFR2/PAK1 axis, thus inhibiting HCC progression [47]. 

Many PAK inhibitors have been developed because of the critical roles of PAKs in various disease conditions. The effects and mechanisms of these inhibitors have been reviewed elsewhere. Although the efficacy of PAK inhibitors has been tested in many preclinical models, none has been successfully tested through a clinical trial. KPT-9274, a PAK4 and nicotinamide phosphoribosyltransferase (NAMPT) dual inhibitor, is the only one under phase I clinical trials (NCT02702492 and NCT04281420) for solid tumors and non-Hodgkin’s lymphoma [114,115,116]. Recently, CP734, a PAK1 inhibitor, was reported to suppress pancreatic cancer growth in mouse [117]. PAKib, a newly identified PAK4 inhibitor, was also found to suppress pancreatic cancer in a mouse model [118]. The efficacy of these newly developed PAK inhibitors in liver disorders are to be tested.

## 8. Discussion

The initiation and progression of liver disorders are accompanied by dysregulation of glucolipid metabolism, as the liver plays a central role in glucose, lipid, and energy metabolism [119]. Therefore, it is necessary to discuss the potential role of PAKs in glucolipid metabolism of the liver to further explore the mechanisms involved in the roles of PAKs in liver disorders. 

Immunotherapy in cancer treatment has achieved great success. The application of immune checkpoint inhibitors has brought significant changes in the management of HCC, which is one of the major disorders in the liver and the 16th leading cause of death worldwide [120]. Thus, it is important to extend our discussion here to the possible effects of PAKs on the immunotherapy of liver disorders.

### 8.1. Potential Role of PAKs in Glucolipid Metabolism of Liver

Compared with normal hepatocytes, glucose transporters are highly expressed in HCC cells for the increased requirement of glucose for sustaining the rapid growth of tumors. PAK1 plays a critical role in glucose production. Aged Pak1^−/−^ mice showed impaired pyruvate tolerance and glucagon-like peptide-1 secretion response, resulting increased glucose disposal in the liver [121]. Fatty acid is another major energy source for HCC cells, especially under poor nutrient supply, and the proteins responsible for fatty acid uptake, transport, and synthesis are upregulated in HCC [122]. AMP-activated protein kinase and mTOR signaling are vital for maintaining the balance of glucolipid metabolism in the liver [123,124]. The activation of the PI3K/AKT/mTOR pathway enhanced lipid synthesis, thereby exacerbating the liver steatosis and promoting HCC progression [125]. The potential role of PAKs in the lipid metabolism of the liver has also been implicated. PAK1 and PAK2 in the liver can be affected by fatty acid and phospholipids. For example, the unsaturated fatty acid oleate was found to stimulate PAK1 activity, and a low-dose cardiolipin also promoted the autophosphorylation of PAK1 and PAK2 [126,127]. The autophosphorylation of PAK2 further facilitated the phosphorylation of histone H2B [127]. PAK4 in fetal hepatocytes was implicated in the sexual dimorphism of metabolic disease through the regulation of de novo lipogenesis [128]. The promoting role of PAK4 in glucose intake and lipid biosynthesis in cancer cells has also been reported [129]. 

Nonalcoholic fatty liver disease (NAFLD) is characterized by abnormal lipid accumulation and comprises fatty liver and non-alcoholic steatohepatitis. NAFLD is the most prevalent liver disorder and recognized as a metabolic syndrome in the liver [130]. NAFLD is one of the primary risk factors for end-stage liver diseases including liver fibrosis/cirrhosis and HCC. Global prevalence of NAFLD is 25.24% and still on the rise, and the affected populations tend to be younger [130,131,132]. Given the role of PAKs in liver fibrosis and HCC, more closed studies are necessary to explore the association between PAKs and NAFLD and the mechanisms involved. Patients with NAFLD are more intolerant to IRI because of the overproduction of reactive oxygen species and disorder of scavenging mechanisms in the liver, thus present with more complications after surgery or liver transplantation [133,134]. IRI occurred in hepatectomy and increased the risk of HCC recurrence in the fatty liver [135]. The utilization of fatty donor liver during transplantation was associated with worse prognosis of patients [136]. Therefore, it is becoming important to examine the role of PAKs in IRI of NAFLD as well as the efficacy of PAKs intervention.

### 8.2. Potential Role of PAKs in Immunotherapy for Liver Disorders

The tumor microenvironment plays key roles in the regulation of immune activation or evasion driven by the opposing actions of antitumor effectors and immunosuppressive components. It is worth noticing that the key cells with immunosuppressive roles implicated in HCC immune evasion include Kupffer cells, monocyte-derived macrophages, regulatory T cells and myeloid-derived suppressive cells [137]. Alteration of PAK1 in Kupffer cells in autoimmune hepatitis implicates the potential immune regulation of PAK in liver diseases [96]. Recently, PAK effects on tumor microenvironment and immune response have been reported. Depletion of PAK1 enhanced the immune system and inhibited intestinal tumorigenesis in a mouse model of intestinal cancer [138]. Inhibition of PAK1 also stimulated the intra-tumoral CD4^+^ and CD8^+^ T cells and suppressed pancreatic cancer [82]. More recently, it has been reported in melanoma that PAK4 expression was elevated in the patients’ samples with poor infiltrated immune cells and thus decreased the response to a PD-1 blockade, and that knockout of PAK4 enhanced the infiltration of CD8^+^ T cells, possibly by downregulation of PD-L1 [139]. These reports indicated the important role of PAKs in tumor immune response and the tumor microenvironment. Although the effects of PAKs on liver cancer have been widely studied, the impact of PAKs on immune response in the tumor microenvironment of liver cancer is rarely reported. The role of PAKs in liver cancer immune microenvironment deserves more investigation.

## 9. Conclusions

Increasing evidence has shown that PAKs play pivotal roles in the progression of liver disorders including liver cancer, hepatic IRI, hepatitis, and liver fibrosis. Elevated PAKs expression in liver cancer stimulates tumor growth and metastasis, contributes to therapeutical resistance, and is associated with the poor prognosis of patients. The studies of the effects of PAK1 and PAK4 in hepatic IRI have provided a useful guide for the potential application of PAK inhibitors in clinical settings. PAK1, the most extensively studied PAKs, plays important roles in the occurrence of hepatitis and liver fibrosis through affecting hepatocytes, immune cells, and myofibroblasts. It is worth noting that PAK1 promotes the development of HBV, hepatitis E virus, parasite, and autoimmune-related hepatitis while inhibiting HCV-induced hepatitis. With the increased occurrence of metabolism-related liver disease such as NAFLD, it is becoming important to explore the association between PAKs and liver metabolism and mechanisms involved. As our knowledge of PAKs and liver disorders accumulates, the efficacy of PAK inhibitors in liver disorders will be explored, which will provide a useful guide in clinical applications.

## Figures and Tables

**Figure 1 cancers-15-00551-f001:**
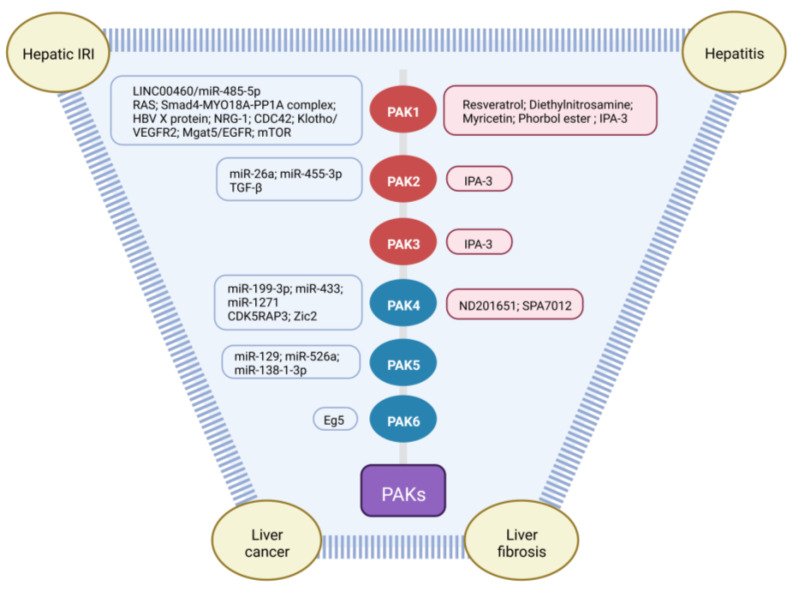
Upstream regulators of PAKs in liver disorders. The upstream regulators of PAKs in liver disorders include exogenous compounds and endogenous molecules (proteins, miRNAs and lncRNAs). CDK5RAP3, CDK5 kinase regulatory subunit-associated protein 3; IRI, ischemia-reperfusion injury; Mgat5, N-acetylglucosaminyltransferase V; PAK, p21 Activated Kinase.

**Figure 2 cancers-15-00551-f002:**
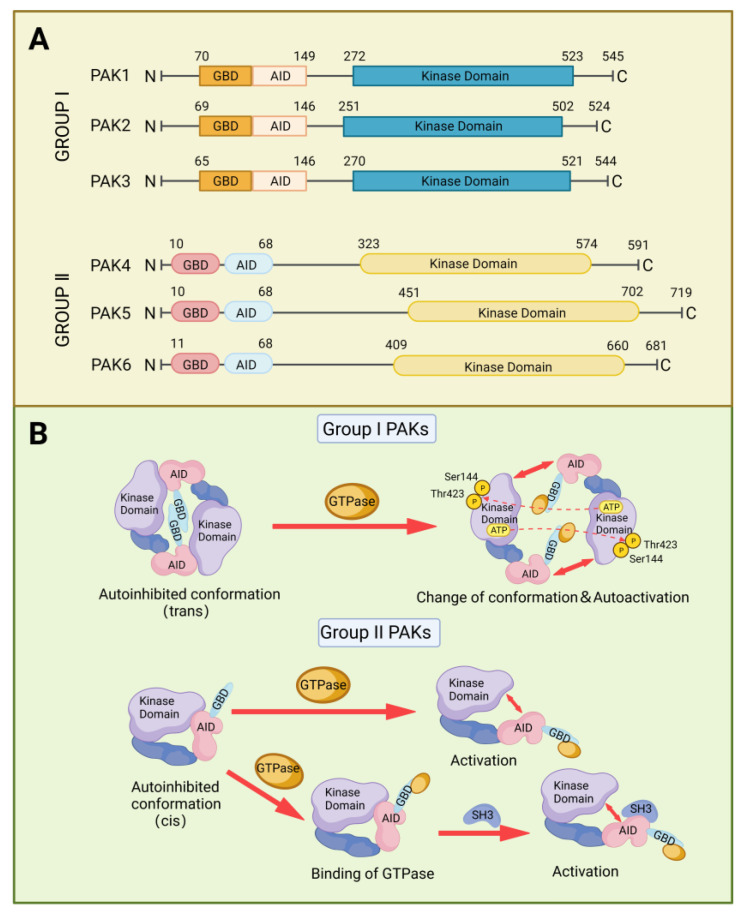
The structure and activation of PAKs. (**A**) Two-dimensional diagram of PAKs structure. (**B**) Different activation mechanisms of Group I PAKs and Group II PAKs.

**Figure 3 cancers-15-00551-f003:**
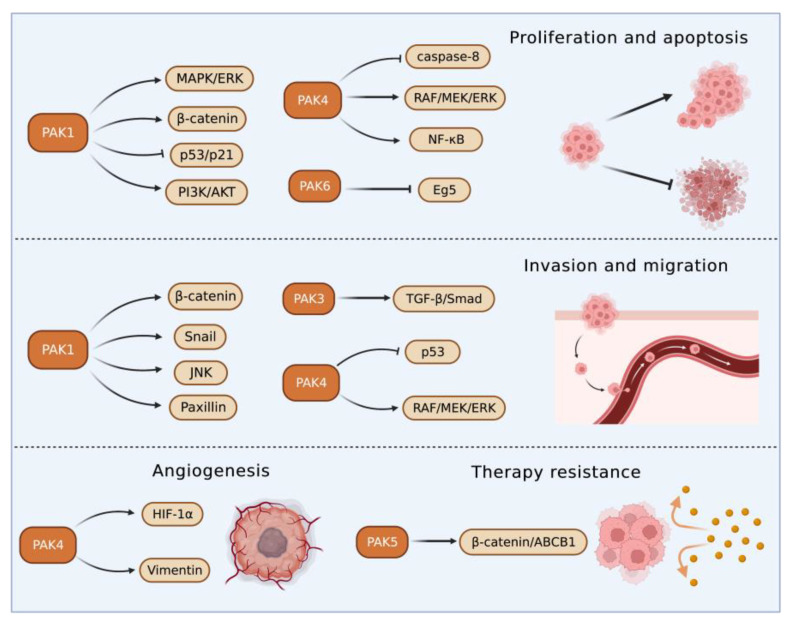
The pathways affected by PAKs in liver cancer. PAKs modulate multiple signaling pathways (e.g., RAF/MEK/ERK, β-catenin) to promote the proliferation, invasion/migration, angiogenesis, and therapy resistance of cancer cells while inhibiting apoptosis.

**Figure 4 cancers-15-00551-f004:**
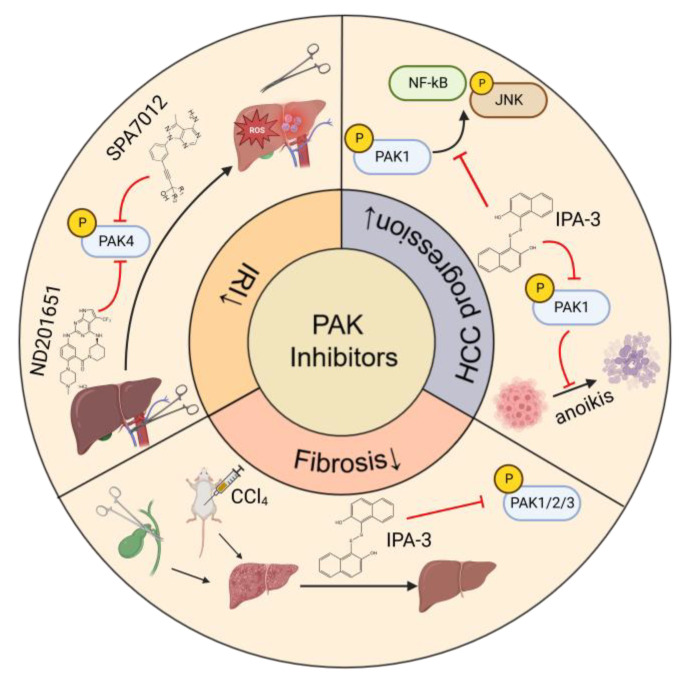
PAK inhibitors applied in liver disorders. Abbreviations: HCC, hepatocellular carcinoma; IRI, ischemia-reperfusion injury; PAK, p21 Activated Kinase; ROS, reactive oxygen species.

**Table 1 cancers-15-00551-t001:** Overview of PAKs in liver disorders.

Liver Disorders	Group I PAKs	Group II PAKs
**Liver cancer**	**PAK1:** β-catenin → Proliferation/invasion↑ and apoptosis↓ MAPK/ERK, PI3K/AKT → Apoptosis↓ Snail → Migration and invasion↑ Snail, β-catenin, p53/p21 → Xenograft tumor growth↑ Paxillin, JNK → Actin stress fibers and metastasis↑ Anchorage-independent growth and anoikis resistance ↑ HIF-1α, Vimentin → Vasculogenic mimicry↑ **PAK2:** Migration↑ **PAK3:** TGF-β/Smad → EMT↑	**PAK4:** Caspase-8, NF-κB → Apoptosis↓ RAF/MEK/ERK → Tumor growth and metastasis↑ P53 → Migration and invasion↑ **PAK5:** Proliferation and invasion↑ Apoptosis induced by cisplatin↓ β-catenin/ABCB1 → Resistance to adriamycin, 5-fluorouracil and sorafenib↑ Radiotherapy resistance↑ **PAK6:** Proliferation↑ Eg5 → mitosis↑ HCC metastasis↑
**Liver IRI**	**PAK1:** VEGFα, IGF1 → IRI↓	**PAK4:** Nrf2 → Inflammation and necrosis of hepatocytes↑
**Hepatitis**	**PAK1:** RAF-1 → HBV X protein → Apoptosis↓ HCV replication↓ NMIIA/Cofilin → HEV infection↑ NF-κB/IRF1/IL-6 → The differentiation of CD4^+^ T cells to Th17 cells and inflammation in *Schistosoma japonicum* infection↑ Autoimmune hepatitis progression↑ **PAK2:** Apoptosis and inflammatory reaction in cells infected with HCV↓ **PAK3:** HBV replication↑	**PAK6:** Tumorigenesis in alcoholic hepatitis↑
**Liver fibrosis**	**PAK1:** Integrin β-1 → Myofibroblasts activation and ECM production↑ **PAK2:** Myofibroblast activation and liver fibrosis↑ **PAK3:** Integrin β-1 → Myofibroblasts activation and ECM production↑	**PAK4:** Perisinusoidal/pericellular fibrosis↑

↑—increment; ↓—decrement.

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
