# Peer review of "P-21 Activated Kinases in Liver Disorders"

_cancers, 2023, doi:10.3390/cancers15020551_

Round 1

Reviewer 1 Report

The current review reads well. Several new Pak inhibitors are shown in literature and this need to be updated

Rest looks fine

Author Response

Please refer to the attached " Address to reviewers' comments"

Reviewer 2 Report

An excellent and comprehensive review, no recommendations

Reviewer 3 Report

In the present review the authors try to provide an overview on current progress of P-21- activated kinases (PAKs) in liver diseases with especial focus on the HCC. However, there are many major issues that should be addressed:

1-      The review is not well organized and need further arrangements, Example: the discussion section ?????

2-       The Table 1 is confusing and need to be more organized. You could arrange the data as follow: Type of PAKs, Liver disorders, Type of change (increase or decrease of the activity or expression, then the affected pathway.

3-      Section 2 (Structure and Activation of PAKs) needs graphical illustration

4-      The pathways affected by PAKs should be under specific heading with illustrating graph for better follow of the data

5-      It is better to add additional sections to the review: One about the physiological role of PAKs (at least in the liver physiology) and another one for the transcriptional and translational regulation of PAKs.

6-      Many abbreviations in the article without the full names

7- The word "Effects" in the subtitles (3. PAKs in liver cancer) should be replaced by (Roles)

Round 2

Reviewer 3 Report

No comment